# Identification of Mean-Field Dynamics using Transformers

## Abstract

This paper investigates the use of transformer architectures to approximate the mean-field dynamics of interacting particle systems exhibiting collective behavior. Such systems are fundamental in modeling phenomena across physics, biology, and engineering, including gas dynamics, opinion formation, biological networks, and swarm robotics. The key characteristic of these systems is that the particles are indistinguishable, leading to permutation-equivariant dynamics. We demonstrate that transformers, which inherently possess permutation equivariance, are well-suited for approximating these dynamics. Specifically, we prove that if a finite-dimensional transformer can effectively approximate the finite-dimensional vector field governing the particle system, then the expected output of this transformer provides a good approximation for the infinite-dimensional mean-field vector field. Leveraging this result, we establish theoretical bounds on the distance between the true mean-field dynamics and those obtained using the transformer. We validate our theoretical findings through numerical simulations on the Cucker-Smale model for flocking, and the mean-field system for training two-layer neural networks.

## 1 Introduction

The identification of dynamical system models for physical processes is a fundamental application of machine learning (ML). Of particular interest are systems of particles exhibiting collective behaviors—such as swarming, flocking, opinion dynamics, and consensus—are of significant interest. These systems involve a large number of particles or agents that follow identical dynamics, which are independent of the particles' identities and are *permutation-equivariant*. Examples include biological entities (Lopez et al., 2012), robots (Elamvazhuthi & Berman, 2019), traffic flow (Piccoli et al., 2009; Siri et al., 2021), and parameters in two-layer neural networks (Mei et al., 2019). A common approach to simplifying the analysis of such systems is to consider the continuum limit as the number of particles $n \to \infty$, resulting in *mean-field models* rooted in statistical physics. Instead of specifying the dynamics of each agent, the particles are modeled using probability measures. This paper learns the mean-field dynamics of particles via particle trajectories using transformers.

There have been several works on learning mean-field dynamics using machine learning. For example, recent works have utilized classical approaches. Pham & Warin (2023) constructed two different types of neural network based function approximators for mean-field mappings and proved their approximation capabilities. The works Feng et al. (2021); Lu et al. (2019) present a kernel-based method for identifying dynamics of interacting particle systems Miller et al. (2023) employed kernel-based methods. The work Messenger & Bortz (2022) presents a weak form of the SINDy algorithm Brunton et al. (2016) for identifying mean-field dynamics of interacting particle systems.

This paper explores the use of transformers to approximate the dynamical systems governing the collective behavior of interacting particles with permutation-equivariant dynamics. We define a new transformer architecture for mean-field or measure-dependent vector fields by taking the expectation of a finite-dimensional transformer with respect to a product measure, which we refer to as the *expected transformer*. This approach differs from recent works Vuckovic et al. (2020); Geshkovski et al. (2023); Furuya et al. (2024a); Adu & Gharesifard (2024) that express transformers as maps on the space of probability measures by defining a continuous version of attention.

Encoding permutation equivariance into the function class or model used for identifying such systems is potentially advantageous due to the benefits of the inductive bias in learning. This raises

the question of which classes of functions guarantee approximation and learning benefits while possessing permutation equivariance. Transformers Vaswani et al. (2017), which have achieved state-of-the-art performance in many learning applications involving sequence-to-sequence mappings, are one such model class.

In this work, we analyze the approximation capabilities of the expected transformer and establish rates of approximation of the expected transformer as a function of the approximation error achieved by the finite-dimensional transformer and the sequence length used. Using this approximation result, we demonstrate that the solution of the *continuity equation*—which describes the mean-field behavior of interacting particle systems—can be approximated by approximating the vector fields using the expected transformer model. We validate our theoretical findings through numerical simulations and comparisons with established benchmarks, specifically on the Cucker-Smale model for swarming, and the mean-field system for training two-layer neural networks.

To summarize, our main **contributions** are as follows:

1. We define a continuum version of the transformer as an expectation of finite-dimensional transformers (see Equation 7). This is distinct from prior work such as Geshkovski et al. (2023) that define neural networks on infinite dimensional spaces directly. Instead, we lift a finite-dimensional model into infinite-dimensional space.
2. We establish approximation rates of measure-valued maps by this expected transformer (see Theorem 3).
3. We show that the solution of the continuity equation can be approximated by approximating the vector fields using the expected transformer (see Theorem 4).

**Other related work**    Universal approximation of functions by neural networks has a long history. One of the prominent earlier results Hornik et al. (1989) proved that continuous functions can be well approximated by using neural networks of bounded depth but arbitrary width. For the standard activation functions, the complementary result for bounded width but arbitrary depth has also been shown in Telgarsky (2016); Yarotsky (2018). For bounded width and depth, Maiorov & Pinkus (1999) provided such a result for special activation functions. Recent results such as Kidger & Lyons (2020); Shen et al. (2022) have investigated bounded width and depth for more standard network architectures. The capabilities of bounded width but arbitrary depth residual networks for approximating solutions of the continuity equation, such as those arising in normalizing flows, has been studied in Ruiz-Balet & Zuazua (2023); Elamvazhuthi et al. (2022).

Significant work has been done using ML to approximate solutions to differential equations, particularly solutions to partial differential equations (PDEs). Specifically, Physics Informed Neural Networks (PINN) were introduced by Raissi et al. (2019) as a method for solving PDEs using neural networks. PINNs have been successfully used in Cai et al. (2021); Weinan & Yu (2017); Bhatnagar et al. (2019). The idea here is to use the differential equation as the loss function for the neural network. Other works have also investigated the problem of approximating the solution operator to PDEs on a mesh (Guo et al., 2016; Zhu & Zabaras, 2018; Adler & Öktem, 2017; Bhatnagar et al., 2019). Building on this, (Kovachki et al., 2021; Li et al., 2020a;b) developed *neural operators* defined on infinite-dimensional spaces to solve PDES, and Furuya et al. (2024b) provide universal approximation results. Additionally, Li et al. (2023) provide quantitative results for approximating eigenfunctions of the Laplace equation on manifolds.

## 2   NOTATION

In this section, we present the notation used throughout the paper. Let $\mathbb{R}^d$ denote the $d$-dimensional Euclidean space, and let $\mathbb{Z}_+$ denote the set of positive integers. The diameter of a subset $A \subset \mathbb{R}^d$ is defined as $\mathrm{diam}(A) := \sup\{\|x - y\| : x, y \in A\}$ and the closed ball of radius $r > 0$ centered at $z \in \mathbb{R}^d$ is denoted by $B_r(z)$.

We denote by $\mathcal{P}(\mathbb{R}^d)$ the set of all Borel probability measures on $\mathbb{R}^d$. The subset of probability measures with finite $p$-th moments is denoted by

$$\mathcal{P}_p(\mathbb{R}^d) := \left\{ \nu \in \mathcal{P}(\mathbb{R}^d) : M_p(\mu) := \left( \int_{\mathbb{R}^d} \|x\|^p \, d\mu(x) \right)^{1/p} < \infty \right\}.$$

We denote by $\mathcal{P}_c(\mathbb{R}^d)$ the set of probability measures with compact support. The set of empirical measures formed by finite sums of $n$ Dirac-delta measures is denoted by

$$\mathcal{D}^n(\mathbb{R}^d) := \left\{ \nu \in \mathcal{P}(\mathbb{R}^d) : \nu = \frac{1}{n} \sum_{i=1}^n \delta_{x_i}, \ x_i \in \mathbb{R}^d \right\}.$$

For $\nu \in \mathcal{P}(\mathbb{R}^d)$, the $n$-fold product measure on $(\mathbb{R}^d)^n$ is denoted by $\nu^{\otimes n} := \underbrace{\nu \times \cdots \times \nu}_{n \text{ times}}$. The support of a measure $\nu \in \mathcal{P}(\mathbb{R}^d)$, denoted by $\mathrm{supp}(\nu)$, is the smallest closed set $S \subset \mathbb{R}^d$ such that $\mu(\mathbb{R}^d \setminus S) = 0$. The $q^{\text{th}}$ moment of a measure $\nu$ is denoted by $M_q(\nu)$. Given a measurable map $X : \mathbb{R}^d \to \mathbb{R}^d$ and a measure $\mu \in \mathcal{P}(\mathbb{R}^d)$, the *pushforward measure* $X_\# \mu \in \mathcal{P}(\mathbb{R}^d)$ is defined by

$$(X_\# \mu)(A) := \mu\left(X^{-1}(A)\right)$$

for every Borel measurable set $A \subset \mathbb{R}^d$.

Boldface letters, such as $\mathbf{z}$, denote elements in $\mathbb{R}^{n \times d}$, representing collections of $n$ vectors in $\mathbb{R}^d$. For a vector $y \in \mathbb{R}^d$, the $i$-th component is denoted by $(y)_i$. We denote by $L^p(\mathbb{R}^d, \mu)$ the space of measurable functions $f : \mathbb{R}^d \to \mathbb{R}$ such that

$$\|f\|_{L^p(\mu)} := \left( \int_{\mathbb{R}^d} |f(x)|^p \, d\mu(x) \right)^{1/p} < \infty.$$

The space of essentially bounded measurable functions is denoted by $L^\infty(\mathbb{R}^d, \mu)$, with the essential supremum norm

$$\|f\|_{L^\infty(\mu)} := \operatorname*{ess\,sup}_{x \in \mathbb{R}^d} |f(x)|.$$

A function $f : (\mathbb{R}^d)^n \to (\mathbb{R}^d)^n$ is *permutation equivariant* if for any permutation $\sigma \in S_n$, where $S_n$ is the symmetric group on $n$ elements, and for any $\mathbf{x} = (x_1, \ldots, x_n) \in (\mathbb{R}^d)^n$, we have

$$f(x_{\sigma(1)}, \ldots, x_{\sigma(n)}) = \left( f_{\sigma(1)}(\mathbf{x}), \ldots, f_{\sigma(n)}(\mathbf{x}) \right),$$

where $f_i(\mathbf{x})$ denotes the $i$-th component of the output. We denote by $C^k(\mathbb{R}^d)$ the space of $k$-times continuously differentiable functions on $\mathbb{R}^d$. The space of continuous functions with compact support is denoted by $C_c(\mathbb{R}^d)$.

## 3 PROBLEM FORMULATION

Let $\Omega \subset \mathbb{R}^d$. Consider a vector field $\mathcal{F} : \Omega \times \mathcal{P}(\Omega) \to \mathbb{R}^d$. The following equation describes the general mean-field behavior of interacting particles evolving on $\Omega$,

$$\frac{dz}{dt} = \mathcal{F}(z, \mu), \ z(0) = z_0 \sim \mu_0, \tag{1}$$

where $z \in \Omega$ denotes the state of a particle, $\mu \in \mathcal{P}(\Omega)$ is the distribution of the particles at time $t$, and $\mu_0$ is the initial distribution. The inter-particle interactions are modeled through $\mu$; specifically, the dynamics of each particle are influenced by the distribution of all particles.

Corresponding to Equation 1, the *continuity equation* describes the evolution of the distribution $\mu$:

$$\frac{\partial \mu}{\partial t} + \nabla_z \cdot (\mathcal{F}(z, \mu)\mu) = 0, \ \mu(0) = \mu_0. \tag{2}$$

For a finite final time $\tau > 0$, we denote by $\mu^{\mathcal{F}} : [0, \tau] \to \mathcal{P}(\Omega)$ the solution of the continuity Equation 2 over the time interval $[0, \tau]$. In this paper, we propose to use transformers to approximate the maps in Equation 1 and Equation 2.

### 3.1 LIFTING TRANSFORMERS TO THE SPACE OF MEASURES

Traditionally, transformers are defined on sequences of vectors in $\mathbb{R}^d$. However, the map we wish to approximate, $\mathcal{F}$, is defined on $\Omega \times \mathcal{P}(\Omega)$. Therefore, we lift the standard transformer to operate on $\Omega \times \mathcal{P}(\Omega)$ via an expectation operation. Before we do this, we briefly review the standard transformer architecture; the following definitions are adapted from Alberti et al. (2023). The core component of the transformer is the multi-head self-attention mechanism.

**Definition 1** (Multi-Headed Self-Attention). *Let $X \in \mathbb{R}^{n \times d}$ be a matrix whose rows are $n$ data points in $\mathbb{R}^d$. Let $W_Q, W_K, W_V \in \mathbb{R}^{d \times d}$ be learnable weight matrices. Define the* query, key, *and* value *matrices by*

$$Q = XW_Q, \quad K = XW_K, \quad V = XW_V.$$

*Let* softmax *denote the softmax function applied row-wise to a matrix. The self-attention head function* AttHead $: \mathbb{R}^{n \times d} \to \mathbb{R}^{n \times d}$ *is defined as*

$$\text{AttHead}(X) := \text{softmax}\left(\frac{QK^\top}{\sqrt{d}}\right) V. \tag{3}$$

*Let $h \in \mathbb{Z}_+$ be the number of attention heads. Let* $\text{AttHead}_1, \dots, \text{AttHead}_h$ *be attention heads with their own weight matrices, and let $W_0 \in \mathbb{R}^{hd \times d}$ be a learnable weight matrix. The multi-head self-attention layer* Att $: \mathbb{R}^{n \times d} \to \mathbb{R}^{n \times d}$ *is defined as*

$$\text{Att}(X) := [\text{AttHead}_1(X), \ \text{AttHead}_2(X), \ \dots, \ \text{AttHead}_h(X)]\, W_0, \tag{4}$$

*where $[\cdot]$ denotes concatenation along the feature dimension.*

**Definition 2** (Transformer Network). *A* transformer block Block $: \mathbb{R}^{n \times d} \to \mathbb{R}^{n \times d}$ *is defined as*

$$\text{Block}(X) := X + \text{FC}\left(X + \text{Att}(X)\right), \tag{5}$$

*where* FC *are feed-forward layers (position-wise fully connected layers), and* ReLU *is the rectified linear unit activation function. The addition operations represent residual connections. Let $L \in \mathbb{Z}_+$, and let* $\text{Block}_1, \dots, \text{Block}_L$ *be transformer blocks. A* transformer network $T : \mathbb{R}^{n \times d} \to \mathbb{R}^{n \times k}$ *is defined as a composition of transformer blocks followed by an output network:*

$$T(X) := \text{FC}_{out}\left(\text{Block}_L \circ \text{Block}_{L-1} \circ \cdots \circ \text{Block}_1(X)\right), \tag{6}$$

*where* $\text{FC}_{out} : \mathbb{R}^{n \times d} \to \mathbb{R}^{n \times k}$ *is a fully connected neural network applied position-wise.*

Next, we introduce the *expected transformer* $\mathcal{T}_n$, which allows us to lift any transformer $T : \Omega^{n+1} \to \mathbb{R}^{(n+1) \times d}$ to a model $\mathcal{T}_n : \Omega \times \mathcal{P}(\Omega) \to \mathbb{R}^d$.

**Definition 3** (Expected Transformer). *Given a transformer $T : \Omega^{n+1} \to \mathbb{R}^{n+1 \times d}$ and a prescribed sequence length $n$, define the* expected transformer $\mathcal{T}_n : \Omega \times \mathcal{P}(\Omega) \to \mathbb{R}^d$ *by*

$$\mathcal{T}_n(x, \mu) := \mathbb{E}_{\mathbf{z} \sim \mu^{\otimes n}}\left[(T([x; \ \mathbf{z}]))_1\right], \tag{7}$$

$$= \int_{\Omega^n} (T([x; \ z_1, \dots, z_n]))_1 \ d\mu(z_1) \cdots d\mu(z_n),$$

*where $[x; \ \mathbf{z}]$ denotes the concatenation of $x$ and $\mathbf{z} = (z_1, \dots, z_n)$ to form an input sequence of length $n+1$, and $(T([x; \ \mathbf{z}]))_1$ denotes the first output vector.*

**Remark 1** (Computing $\mathcal{T}_n(x, \mu)$). *Given a finite dimensional transformer $T$, $\mathcal{T}_n$ can be approximated empirically. In particular, let the data tensor be of size $B \times n \times d$, where $B$ is the batch size, and the sequence size $n \times d$ are sampled from $\mu^{\otimes n}$. Then, given a point, $x$, we add it to each sequence, process the whole batch at once, and compute the mean. In this manner, inference with the expected transformer is straightforward.*

Some prior works Geshkovski et al. (2023); Furuya et al. (2024a) have defined transformers $\hat{T} : \Omega \times \mathcal{P}(\Omega) \to \mathbb{R}^d$ through a continuous version of self-attention $\Gamma$. For $x \in \Omega$ and $\mu \in \mathcal{P}(\Omega)$, $\Gamma$ is defined as

$$\Gamma(x, \mu) := x + \frac{1}{Z(x, \mu)} \int_\Omega \text{Att}([x; \ y]) \ d\mu(y), \tag{8}$$

where $Z(x, \mu)$ is a normalization factor. Then the transformer $\hat{T}$ in Geshkovski et al. (2023); Furuya et al. (2024a) is defined as

$$\hat{T}(x, \mu) := \text{FC}_{\xi_L} \circ \Gamma_{\theta_L} \circ \cdots \circ \text{FC}_{\xi_1} \circ \Gamma_{\theta_1}(x), \tag{9}$$

where $\Gamma_{\theta_j}$ and $\text{FC}_{\xi_j}$ are attention and feed-forward layers with parameters $\theta_j$ and $\xi_j$, respectively. When $\mu$ is an empirical measure (a sum of Dirac deltas), this formulation reduces to the standard transformer definition 6. However, due to the nested expectations, computing, or even approximating, $\hat{T}(x, \mu)$ is not straightforward.

**Goals** Our objectives are twofold. First, in Theorem 3, we show that, given a vector field $\mathcal{F}$ as in Equation 1, we can approximate it by the expected transformer $\mathcal{T}_n$ as defined in Equation 7 in a suitable sense. Second, we wish to approximate the solution of the continuity equation 2. Towards this goal, we define an approximate continuity equation using $\mathcal{T}_n$:

$$\frac{\partial \mu}{\partial t} + \nabla_z \cdot (\mathcal{T}_n(z, \mu)\mu) = 0, \quad \mu(0) = \mu_0. \tag{10}$$

Let $\mu^{\mathcal{F}}(t)$ be the solution to Equation 2, and let $\mu^{\mathcal{T}_n}(t)$ be the solution to Equation 10. In Theorem 4, we will prove that $\mu^{\mathcal{T}_n}(t)$ approximates $\mu^{\mathcal{F}}(t)$ in a suitable sense.

### 3.2 Universal Approximation of Transformers

At the heart of our argument is the approximation result for finite dimensional transformers can be lifted to approximation results for the expected transformer. Universal approximation of functions on $\mathbb{R}^{d \times n}$ by transformers was first proved in Yun et al. (2020). The approximation was proved under the $L^p(\mathbb{R}^{d \times n})$ norm. The recent work by Alberti et al. (2023), stated below, improves this prior result by proving approximation under the uniform norm.

**Theorem 1** (Universal Approximation by Transformer Alberti et al. (2023)). *Let $f : \mathbb{R}^{d \times n} \to \mathbb{R}^{d \times n}$ be a permutation equivariant function, for each $\varepsilon < 0$, there exists a transformer $T$ such that,*

$$\sup_{X \in \mathbb{R}^{d \times n}} \|f(X) - T(X)\|_\infty < \varepsilon.$$

The concurrent work Furuya et al. (2024a) proves the following approximation result for continuous maps $\mathcal{F}$ by the continuum version of the transformer $\hat{T}$ (equation 9).

**Theorem 2.** *Let $\Omega \subset \mathbb{R}^d$ be a compact set and $F^* : \Omega \times \mathcal{P}(\mathbb{R}^d) \to \mathbb{R}^d$ be continuous, where $\mathcal{P}(\mathbb{R}^d)$ is endowed with the weak\* topology. Then for all $\varepsilon > 0$, there exist $l$ and parameters $(\theta_j, \xi_j)_{j=1}^l$ such that*

$$\|\hat{T}(\mu, x) - F^*(\mu, x)\| \le \varepsilon, \quad \forall (\mu, x) \in \Omega \subset \mathcal{P}(\mathbb{R}^d)$$

*where the parameters $\theta_j, \xi_j$ depend linearly on the dimension $d$.*

## 4 Theoretical Results

To state the result, we require some assumptions on the map $\mathcal{F}$. One key assumption we make is that $\mathcal{F}$ is Lipschitz continuous with respect to its second argument, the probability measure. To formalize this, we require a metric on the space of probability measures $\mathcal{P}(\Omega)$. A commonly used metric is the $p$-Wasserstein distance.

**Definition 4** ($p$-Wasserstein Distance). *Given two probability measures $\mu, \nu \in \mathcal{P}_p(\Omega)$ on a metric space $(\Omega, d)$, where $d$ is the metric on $\Omega$, the $p$-Wasserstein distance between $\mu$ and $\nu$ is defined as*

$$\mathcal{W}_p(\mu, \nu) := \left( \inf_{\gamma \in \Pi(\mu, \nu)} \int_{\Omega \times \Omega} d(x, y)^p, d\gamma(x, y) \right)^{1/p}, \tag{11}$$

*where $\Pi(\mu, \nu)$ denotes the set of all couplings (transport plans) $\gamma$ on $\Omega \times \Omega$ with marginals $\mu$ and $\nu$.*

When $\Omega$ is compact, the $p$-Wasserstein distance metrizes the weak convergence of probability measures ( Theorem 6.9 in Villani (2008)). That is, if $\mu_n \in \mathcal{P}_p(\Omega)$ is a sequence of measures such that $\mu_n \to \mu$ weakly if and only if

$$\mathcal{W}_p(\mu_n, \mu) \to 0.$$

Additionally, since $\Omega$ is compact, convergence in $\mathcal{W}_p$ implies convergence in $\mathcal{W}_q$, for all $q < p$.

We now state the main assumptions required for our analysis.

**Assumption 1** (Regularity and Growth Conditions). *Assume that the vector field $\mathcal{F} : \Omega \times \mathcal{P}_p(\Omega) \to \mathbb{R}^d$ satisfies the following conditions:*

a) *(**Lipschitz Continuity**) There exists a constant $\mathscr{L}$ such that for all $x, y \in \Omega$ and $\mu, \nu \in \mathcal{P}_p(\Omega)$,*

$$\|\mathcal{F}(x, \mu) - \mathcal{F}(y, \nu)\|_{L_\infty} \le \mathscr{L} \left( \|x - y\|_{L_\infty} + \mathcal{W}_p(\mu, \nu) \right),$$

b) (**Linear Growth**) *There exists a constant $\mathscr{M} > 0$ such that for all $x \in \Omega$ and $\mu \in \mathcal{P}_p(\Omega)$,*

$$\|\mathcal{F}(x,\mu)\|_{L_1} \leq \mathscr{M}\left(1 + \|x\|_{L_1} + M_1(\mu)\right),$$

*where $M_1(\mu) := \int_\Omega |y| d\mu(y)$ is the first moment of $\mu$.*

c) (**Smoothness**) *For each $\mu \in \mathcal{P}_p(\Omega)$, the function $x \mapsto \mathcal{F}(x,\mu)$ is continuously differentiable on $\Omega$; that is, $\mathcal{F}(\cdot,\mu) \in C^1(\Omega)$.*

**Remark 2.** *Note that Assumption 1*a) *implies Assumption 1*b).

These assumptions are standard in the analysis of mean-field models and are satisfied by many classical systems, such as the Cucker-Smale flocking model (Cucker & Smale, 2007). Theorem 2 of Piccoli et al. (2015) shows that the model satisfies the conditions of Assumption 1. In Section 5, we provide numerical simulations based on this model. To establish our approximation results for functions $\mathcal{H} : \Omega \times \mathcal{P}(\Omega) \to \mathbb{R}^d$, we define the following norm.

**Definition 5.** *Given a function $\mathcal{H} : \Omega \times \mathcal{P}(\Omega) \to \mathbb{R}^d$, we define its norm by*

$$\|\mathcal{H}\| := \sup x \in \Omega \sup_{\mu \in \mathcal{P}(\Omega)} \|\mathcal{H}(x,\mu)\|_{L_\infty}. \tag{12}$$

### 4.1 Approximating the Vector Field $\mathcal{F}$

We begin by considering the finite-dimensional approximation of the model (equation 1) via a particle-level system defined on $\Omega$. Specifically, let the state of each particle $i \in \{1,\ldots,n\}$ be represented by $z_i \in \Omega$. We assume that each $z_i$ is independently sampled from the distribution $\mu \in \mathcal{P}(\Omega)$. Let $\mathbf{z} = (z_1,\ldots,z_n) \in \Omega^n$ denote the collection of particle states. The empirical distribution of the $n$-particle system is then given by

$$\nu_{\mathbf{z}}^n := \frac{1}{n}\sum_{i=1}^n \delta_{z_i} \in \mathcal{D}^n(\Omega). \tag{13}$$

The particle-level dynamics on $\mathbb{R}^d$ according to the map defined in equation 1 can be written as

$$\dot{z}_i = \mathcal{F}(z_i, \nu_{\mathbf{z}}^n). \tag{14}$$

Note that the collection of random variables $(z_i)$ is permutation equivariant because the joint distribution of $(z_i)$ is invariant under any permutation of the indices. To approximate the vector field $\mathcal{F}$, we define, for a fixed $n$, the finite-dimensional map $F_n : \Omega^n \to \mathbb{R}^{d \times n}$ as

$$F_n(\mathbf{z}) := [\mathcal{F}(z_1, \nu_{\mathbf{z}}^n) \quad \ldots \quad \mathcal{F}(z_n, \nu_{\mathbf{z}}^n)] \tag{15}$$

We now state our main result regarding the universal approximation of the mean-field vector field $\mathcal{F}$ by the expected transformer $\mathcal{T}_n$.

**Theorem 3** (Universal Approximation). *Let $\varepsilon > 0$. Let $\Omega \subset \mathbb{R}^d$ be a compact set containing $0$. Let $\mathcal{F} : \Omega \times \mathcal{P}(\Omega) \to \mathbb{R}^d$ satisfy Assumption 1*a) *for a given $p$. Suppose that there exists a transformer network $T : \Omega^{n+1} \to \mathbb{R}^{(n+1) \times d}$ such that*

$$\sup_{z_1,\ldots,z_{n+1} \in \Omega} \|T(z_1,\ldots,z_{n+1}) - F_{n+1}(z_1,\ldots,z_{n+1})\|_\infty \leq \varepsilon. \tag{16}$$

*Then, for any $q > p$ there exists a constant $C(p,q,d)$, depending only on $p$, $q$, and $d$, such that for all $n \geq 1$, the corresponding continuum version $\mathcal{T}_n : \Omega \times \mathcal{P}(\Omega) \to \mathbb{R}^d$ (equation 7) satisfies*

$$\|\mathcal{T}_n - \mathcal{F}\| \leq \varepsilon + C\mathscr{L}\, diam(\Omega)^p \left(\frac{1}{n^{\frac{q-p}{q}}} + \begin{cases} n^{-1/2} & p > d/2,\ q \neq 2p \\ n^{-1/2}\log(n+1) & p = d/2,\ q \neq 2p \\ n^{-p/d} & p < d/2,\ q \neq d/(d-p) \end{cases}\right).$$

*Proof Sketch.* The main steps of the proof are as follows. For a given measure $\mu \in \mathcal{P}(\Omega)$, consider its empirical approximation $\nu_{\mathbf{z}}^n$, where $\mathbf{z} = (z_1,\ldots,z_n)$ are i.i.d. samples from $\mu$. First, using Assumption 1a), we show that for any $x \in \Omega$, the difference $\|\mathcal{F}(x,\mu) - \mathcal{F}(x,\nu_{\mathbf{z}}^n)\|$ can be bounded in terms of $\mathcal{W}_p(\mu, \nu_{\mathbf{z}}^n)$. We then bound the distance between $\mu$ and $\nu_{\mathbf{z}}^n$ by using Theorem 1 from Fournier & Guillin (2015).

Second, we define the finite-dimensional map $F_{n+1}(x, \mathbf{z})$ as Equation 15, and using the Lipschitz continuity of $\mathcal{F}$, show that the first component $(F_{n+1}(x, \mathbf{z}))_1$ approximates $\mathcal{F}(x, \nu^n \mathbf{z})$ well.

Third, since the transformer $T$ approximates $F_{n+1}$ uniformly within $\varepsilon$, we conclude that $\mathcal{T}_n(x, \mu)$, which is defined as the expected value of $(T(x, \mathbf{z}))_1$ over $\mathbf{z} \sim \mu^{\otimes n}$, approximates $\mathcal{F}(x, \mu)$ within the stated bound. $\square$

In Theorem 3, we have shown that finite-dimensional transformers can approximate maps on infinite-dimensional spaces under the uniform norm (Equation 12). In particular, we see that the approximation depends on two quantities. First, the approximation depends on how well the finite dimensional transformer $T$ approximates our finite dimensional map $F_n$, which itself is an approximation of $\mathcal{F}$. This corresponds to the $\varepsilon$ term in the bound. In particular, any universal approximation rates for transformers instantly lifts to expect transformer.

Second it depends on the convergence rates of $\mathcal{W}_p(\mu, \nu_{\mathbf{z}}^n)$. We obtain these rates from Theorem 1 of Fournier & Guillin (2015), which depend on $n$, $p$, $q$, and $d$. Fournier & Guillin (2015) showed that these rates are tight. Furthermore, we see that the stronger regularity the map $\mathcal{F}$ has, i.e., the larger the value of $p$, the easier it is to approximate. The best rates are obtained for $p = \lfloor \frac{d}{2} + 1 \rfloor$. Thus, if we better approximate $F_n$ or use longer sequences, we obtain an improved approximation of the vector-field $\mathcal{F}$.

**Comparison with Result From Furuya et al. (2024a):**    To compare with Theorem 2, we state the following corollary to Theorem 3.

**Corollary 1.** *Let $\varepsilon > 0$ and $n \geq 1$. Let $\Omega \subset \mathbb{R}^d$ be a compact set containing $0$. Let $\mathcal{F} : \Omega \times \mathcal{P}(\Omega) \to \mathbb{R}^d$ satisfy Assumption 1a) for a given $p$. Then there exists a transformer $T$ with depth $\Theta(1)$, one attention layer with width $\Theta(d)$ such that the expected transformer $\mathcal{T}_n$ satisfies*

$$\|\mathcal{T}_n - \mathcal{F}\| \leq \varepsilon + C\mathscr{L} \, diam(\Omega)^p \left( \frac{1}{n^{\frac{q-p}{q}}} + \begin{cases} n^{-1/2} & p > d/2, \, q \neq 2p \\ n^{-1/2} \log(n+1) & p = d/2, \, q \neq 2p \\ n^{-p/d} & p < d/2, \, q \neq d/(d-p) \end{cases} \right).$$

*Proof.* We use Theorem 4.3 from Alberti et al. (2023) to construct the transformer $T$ that satisfies the assumptions for Theorem 3. $\square$

While Corollary 1 and Theorem 2 are about two different models, they share notable similarities while exhibiting key differences. Both results feature $\Theta(d)$ width for the attention layers, independent of $\varepsilon$ and $n$, and neither provides bounds for the width of feedforward layers. However, our work establishes a bound on network depth, which Furuya et al. (2024a) does not. Moreover, we note that providing a bound on the feedforward network, in our case, is straightforward, owing to recent developments that provide bounds on both width and depth Augustine (2024). Lastly, we note that Furuya et al. (2024a) impose weaker assumptions for the map $\mathcal{F}$.

### 4.2 Approximating the Mean Field Dynamics

In this section, we build upon our previous approximation results to show that solutions of the continuity equation 2 can be approximated by approximating the vector field $\mathcal{F}$ using a transformer $\mathcal{T}_n$. To formalize this, we first introduce an appropriate notion of a solution to the continuity equation.

**Definition 6.** *A measure-valued function $\mu \in C([0, \tau]; \mathcal{P}_p(\mathbb{R}^d))$ is called a* Lagrangian solution *of the continuity equation 2 if there exists $X : [0, \tau] \times \mathbb{R}^d \to \mathbb{R}^d$, referred to as the flow map, satisfying*

$$X(t, x) = x + \int_0^t \mathcal{F}(X(s, x), \mu(s))ds \tag{17}$$

*for all $x \in \mathbb{R}^d$ and $\mu(t) = X(t, \cdot)_{\#}\mu_0$ for all $t \in [0, \tau]$.*

Under Assumption 1a), it is known that there is a unique Lagrangian solution corresponding to 2. See Proposition 4.8 in Cavagnari et al. (2022). However, a transformer might not be globally Lipschitz as required in Assumption 1a). Hence, the transformer continuity equation 10 may not have a unique Lagrangian solution. For this reason we also need the following assumption.

**Assumption 2.** *The vector field $\mathcal{T}_n$ is such that there exists a unique Lagrangian solution $\mu^{\mathcal{T}_n} \in C([0, \tau]; \mathcal{P}_p(\mathbb{R}^d))$ to the continuity equation*

$$\frac{\partial \mu}{\partial t} + \nabla_z \cdot (\mathcal{T}_n(z, \mu)\mu) = 0, \tag{18}$$

*with initial condition $\mu(0) = \mu_0$.*

We are now ready to state our main theorem regarding the approximation of mean-field dynamics using transformers.

**Theorem 4** (Mean Field Dynamics Approximation Using Transformers). *Let $\delta > 0$ and $n \geq 1$. Suppose $\mathcal{F}$ satisfies Assumption 1 for some $p$ and Assumption 2 for $\mu_0 \in \mathcal{P}_c(\mathbb{R}^d)$. If $\text{supp}\,\mu_0 \subseteq B_R(0)$ for some $R > 0$, and $K \in \mathbb{R}$ is such that $K > (R + 2\mathcal{M}\tau e^{3(\mathcal{M}\tau)})$. $\mathcal{T}_n$ satisfies for all $z \in \bar{B}_K(0)$ and $\mu \in \mathcal{P}(\bar{B}_K(0))$*

$$\|\mathcal{T}_n(x, \mu) - \mathcal{F}(z, \mu)\|_{L_\infty} < \delta$$

*Then we have that*

$$W_p(\mu^{\mathcal{F}}(t), \mu^{\mathcal{T}_n}(t)) < \delta \cdot 2^p t \exp(2^p \mathcal{L} t) \tag{19}$$

*where $\mu^{\mathcal{F}}$ and $\mu^{\mathcal{T}_n}$ are the solutions to equation 2 and equation 10, respectively, and the constants are independent of $\mu_0 \in \mathcal{P}(B_R(0))$.*

Theorem 4 shows that if $\mathcal{T}_n$ approximates $\mathcal{F}$ well then we can use $\mathcal{T}_n$ to simulate the dynamics equation 2 over any time interval. We observe that the error bound 19 grows exponentially. Therefore, small approximation error for the vector field implies small approximation error for the solution of the continuity equation only for small time horizons. However, we note that bound also depends on the regularity of $\mathcal{F}$, namely $p$ and $\mathcal{L}$. Therefore, the more regular the vector-field $\mathcal{F}$, i.e., larger $p$ and smaller $\mathcal{L}$, the better the bound 19.

We can combine Theorem 3 and Theorem 4 to obtain the following corollary.

**Corollary 2.** *Suppose $\mathcal{F}$ satisfies Assumption 1 for some $p$ and Assumption 2 for $\mu_0 \in \mathcal{P}_c(\mathbb{R}^d)$. If $\text{supp}\,\mu_0 \subseteq B_R(0)$ for some $R > 0$, and $K \in \mathbb{R}$ is such that $K > (R + 2\mathcal{M}\tau e^{3(\mathcal{M}\tau)})$. Then, for all $\varepsilon > 0$ small enough and $n \in \mathbb{Z}_+$ large enough, there exists a transformer network $T : \mathbb{R}^{d \times (n+1)} \to \mathbb{R}^{d \times (n+1)}$, with its corresponding continuum version $\mathcal{T}_n : \mathbb{R}^d \times \mathcal{P}(\mathbb{R}^d) \to \mathbb{R}^d$ such that*

$$W_p(\mu^{\mathcal{F}}(t), \mu^{\mathcal{T}_n}(t)) < 2^p(\varepsilon + \delta(n, K)) t \exp(2^p \mathcal{L} t)$$

*where*

$$\delta(n, K) = \mathcal{L}(2K)^p \left( \frac{1}{n^{\frac{q-p}{q}}} + \begin{cases} n^{-1/2} & p > d/2,\, q \neq 2p \\ n^{-1/2}\log(n+1) & p = d/2,\, q \neq 2p \\ n^{-p/d} & p < d/2,\, q \neq d/(d-p) \end{cases} \right).$$

## 5 NUMERICAL SIMULATIONS

While Corollary 1 establishes the existence of a transformer network that approximates the vector field, it does not provide a method for determining the model weights. This section presents experiments where we train transformers to approximate the finite-dimensional vector fields and then use them to simulate solutions. To proceed, suppose that we have $N$ training data points that correspond to input-output pairs $\{(\nu_{\mathbf{z}}^n)^{(j)}, (F_n(\mathbf{z}))^{(j)}\}_{j=1,\dots,N}$. We train the transformer $T$ on this data using the mean squared loss. We will consider two examples of mean-field systems. The first is a synthetic example where we construct the data from the Cucker-Smale (Cucker & Smale, 2007). The second is the mean-field dynamics of training two-layer neural networks (Mei et al., 2019).

**Cucker-Smale** The first example we consider is the well-studied 2-dimensional Cucker-Smale equation that models consensus of a $N$-agent system Cucker & Smale (2007). In the equation below, $x \in \mathbb{R}^2$ and $v \in \mathbb{R}^2$ denote the position and velocity of each agent, respectively. Hence, in this setup, $d = 4$. The vector field $\mathcal{F} : \mathbb{R}^4 \times \mathcal{P}(\mathbb{R}^4) \to \mathbb{R}^4$ is given by

$$\mathcal{F}(x, v, \mu) = \begin{bmatrix} v \\ -\int_{\mathbb{R}^4} \phi(\|x - y\|)(v - u)d\mu(y, u) \end{bmatrix}, \quad \phi(r) = \frac{H}{(s^2 + r^2)^b}.$$

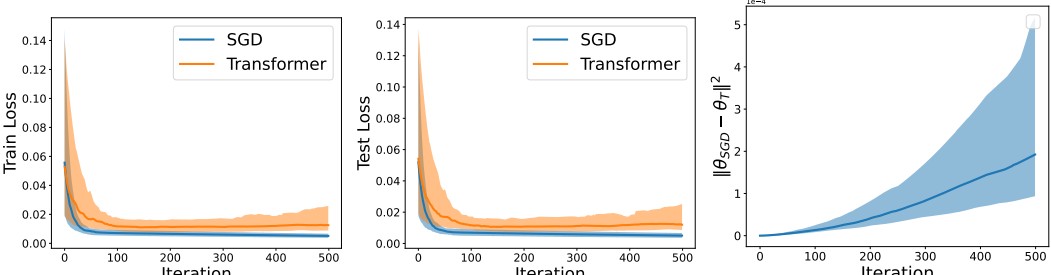

Figure 1: Figure comparing training a two-layer neural network using gradient descent to update the weights and using a transformer to update the weights. The solid line is the median value over 100 trials, while the shaded region is the interquartile range (25th-75th percentile). Left: evolution of the training error during training. Center: evolution of the test error during training. Right: difference between the parameters learned by gradient descent and the transformer.

Here, $\phi$, a non-negative function, is the interaction potential that determines the inter-agent interaction, and $H, s, b$ are parameters (here set to 1). We also consider the particle version of the system.

$$\frac{dx_i}{dt} = v_i, \ 0 \le i \le N \tag{20}$$

$$\frac{dv_i}{dt} = \frac{1}{N} \sum_{j=1}^{N} \phi(\|x_i - x_j\|)(v_j - v_i) \tag{21}$$

To generate the data, we compute the trajectory for 500 different initial conditions. Each initial condition $(x_0^{(j)}, v_0^{(j)})$ is chosen uniformly at random from $\Omega = [0,1] \times [0,1]$. For each initial condition, we generate solution trajectories for $N = 20$ agents over a time horizon $[0, 100]$ using SciPy's `solve_ivp` using the BDF method. Hence, for each initial condition, we get $T_j$ time steps $t_1^{(j)}, \ldots, t_{T_j}^{(j)}$ and for each time step, the method gives us the position and velocities at those time points. Hence, we use a transformer to approximate the $\begin{bmatrix} x \\ v \end{bmatrix} \mapsto \begin{bmatrix} \dot{x} \\ \dot{v} \end{bmatrix}$. We need to compute $\dot{v}$ for each training point. We could have used the true equation to simulate $\dot{v}$, but to align better with real-world scenarios, we compute $\dot{v}$ using the centered difference method, ignoring the points at the boundary. This gives a total of 16834 data points.

**Training 2-Layer Neural Network**    Consider a two-layer network $f(x) = \sum_{i=1}^{N} a_i \sigma(x^T w_i)$. Let $\theta_i = (a_i, w_i)$ be the parameters. In this model, we consider each $\theta_i$ as a particle, and its distribution evolves as we train the model. Mei et al. (2019) showed that the following continuity equation can model the dynamics in the $N \to \infty$ limit.

$$\dot{\mu} = 2\xi(t)\nabla_\theta \cdot (\mu \nabla_\theta \Psi(\theta, \rho)),$$

where $\xi(t)$ depends on the learning rate schedule and

$$\Phi(\theta) = (a, w), \mu) := -\mathbb{E}_{x,y}\left[ya\sigma(x^T w)\right] + \int \mathbb{E}_{x,y}\left[a\hat{a}\sigma(x^T w)\sigma(x^T \hat{w})\right] d\mu(\hat{a}, \hat{w}).$$

Here, the vector field we wish to approximate by the transformer is $\mathcal{F}(\theta, \mu) = 2\xi(t)\nabla_\theta \Psi(\theta, \mu)$. To generate data, we fix a two-layer teacher network with sigmoid activation and use isotropic Gaussian inputs. We set $N = 100$ and use an input dimension of 10. Since the sigmoid is 1-Lipschitz, the above equations satisfy our assumptions.

## 5.1 RESULTS

Figure 1 illustrates the training and test loss as we train the model. The blue line represents

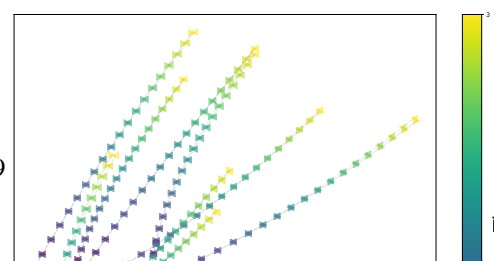

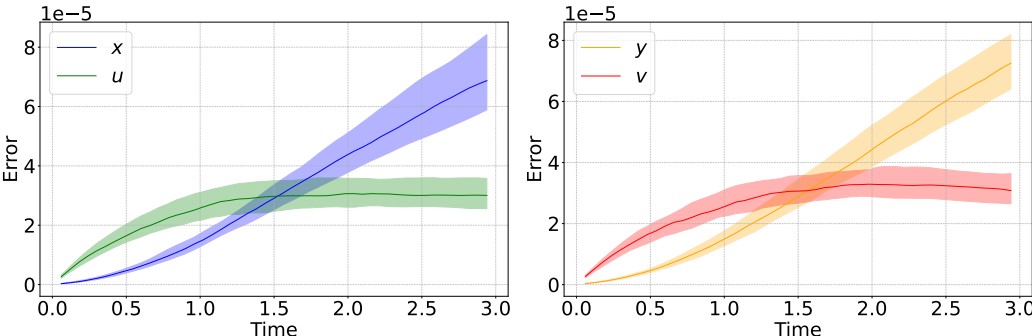

Figure 2: Figure comparing the true dynamics of the Cucker-Smaler versus those obtained from a transformer. The solid line is the median value over 100 trials, while the shaded region is the interquartile range (25th-75th percentile).

the model trained using SGD, while the orange line corresponds to the model trained with the transformer. Note that the transformer model does not compute any gradients. We trained models with one hundred different random initializations. The solid lines indicate the median, and the shaded regions represent the interquartile range. The transformer-trained models exhibit favorable training and test loss performance. The rightmost plot in Figure 1 shows the Frobenius norm of the difference between the parameters learned using SGD and the transformer. The figure demonstrates that the maximum norm of the difference is at most $5 \times 10^{-5}$, even after 500 iterations.

Next, we simulate the Cucker-Smale flocking dynamics. Each particle is represented by a point in two-dimensional space. Figure 3 shows the evolution of ten particles using the Cucker-Smale equations equation 20-equation 21 along with the transformer approximation. Notably, the transformer tracks the true solution quite well. Figure 2 plots the $L_2$ distance between the positions coordinates $x, y$ and velocities $u, v$. The figure indicates that the error is generally quite small ($< 10^{-4}$), although it increases over time. This increase appears to be linear for the position coordinates, while the error in the velocity seems to plateau and even decrease slightly. Additionally, the initial interquartile range is small, but it grows over time.

## 6 CONCLUSION

In conclusion, this paper demonstrated the efficacy of transformer architectures in approximating the mean-field dynamics of interacting particle systems. We showed that finite-dimensional transformer models can be lifted to approximate the infinite-dimensional mean-field dynamics. Through theoretical results on approximations of the vector field and solution to the continuity equation, as well as numerical simulations, we established that transformers can be powerful tools for modeling and learning the collective behavior of particle systems. In the future, we would like to investigate if transformers can be used to for mean-field control.

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

## A  PROOF OF THEOREM 3

**Theorem 3** (Universal Approximation). *Let $\varepsilon > 0$. Let $\Omega \subset \mathbb{R}^d$ be a compact set containing $0$. Let $\mathcal{F} : \Omega \times \mathcal{P}(\Omega) \to \mathbb{R}^d$ satisfy Assumption 1a) for a given $p$. Suppose that there exists a transformer network $T : \Omega^{n+1} \to \mathbb{R}^{(n+1) \times d}$ such that*

$$\sup_{z_1, \dots, z_{n+1} \in \Omega} \|T(z_1, \dots, z_{n+1}) - F_{n+1}(z_1, \dots, z_{n+1})\|_\infty \leq \varepsilon. \tag{16}$$

*Then, for any $q > p$ there exists a constant $C(p, q, d)$, depending only on $p$, $q$, and $d$, such that for all $n \geq 1$, the corresponding continuum version $\mathcal{T}_n : \Omega \times \mathcal{P}(\Omega) \to \mathbb{R}^d$ (equation 7) satisfies*

$$\|\mathcal{T}_n - \mathcal{F}\| \leq \varepsilon + C\mathscr{L} \, diam(\Omega)^p \left( \frac{1}{n^{\frac{q-p}{q}}} + \begin{cases} n^{-1/2} & p > d/2, \, q \neq 2p \\ n^{-1/2} \log(n+1) & p = d/2, \, q \neq 2p \\ n^{-p/d} & p < d/2, \, q \neq d/(d-p) \end{cases} \right).$$

*Proof.*

$$\|\mathcal{F} - \mathcal{T}_n\| = \sup_\mu \sup_x \left\| \mathcal{F}(x, \mu) - \int_{\Omega^n} (T(x, \mathbf{z}))_1 \, d\mu^{\otimes n}(\mathbf{z}) \right\|_\infty$$

$$\leq \sup_\mu \sup_x \int_{\Omega^n} \|\mathcal{F}(x, \mu) - (T(x, \mathbf{z}))_1\|_\infty \, d\mu^{\otimes n}(\mathbf{z})$$

$$= \sup_\mu \sup_x \int_{\Omega^n} \|\mathcal{F}(x, \mu) - \mathcal{F}(x, \nu_{\mathbf{z}}^n) + \mathcal{F}(x, \nu_{\mathbf{z}}^n) - (T(x, \mathbf{z}))_1\|_\infty \, d\mu^{\otimes n}(\mathbf{z})$$

$$\leq \sup_\mu \sup_x \int_{\Omega^n} \|\mathcal{F}(x, \mu) - \mathcal{F}(x, \nu_{\mathbf{z}}^n)\|_\infty \, d\mu^{\otimes n}(\mathbf{z})$$

$$+ \sup_\mu \sup_x \int_{\Omega^n} \|\mathcal{F}(x, \nu_{\mathbf{z}}^n) - (T(x, \mathbf{z}))_1\|_\infty \, d\mu^{\otimes n}(\mathbf{z}) \qquad (*)$$

The second inequality follows from the standard triangle inequality. The first integral on the RHS can be bounded from above as,

$$\sup_\mu \sup_x \int_{\Omega^n} \|\mathcal{F}(x, \mu) - \mathcal{F}(x, \nu_{\mathbf{z}}^n)\|_\infty \, d\mu^{\otimes n}(\mathbf{z})$$

$$\leq \sup_\mu \int_{\Omega^n} \sup_x \|\mathcal{F}(x, \mu) - \mathcal{F}(x, \nu_{\mathbf{z}}^n)\|_\infty \, d\mu^{\otimes n}(\mathbf{z})$$

$$= \sup_\mu \int_{\Omega^n} \mathscr{L} \|\mu - \nu_{\mathbf{z}}^n\|_{\mathcal{W}_p} \, d\mu^{\otimes n}(\mathbf{z})$$

$$= \mathscr{L} \sup_\mu \mathbb{E}_{\mathbf{z} \sim \mu^{\otimes n}} [\mathcal{W}_p(\mu, \nu_{\mathbf{z}}^n)] \qquad (**)$$

We let $M_q(\mu)$ be the $q$-moment of $\mu$ i.e. $M_q(\mu) := \int_\Omega |x|^q d\mu(x)$ and

$$G(n) = \begin{cases} n^{-1/2} & p > d/2, \, q \neq 2p \\ n^{-1/2} \log(n+1) & p = d/2, \, q \neq 2p \\ n^{-p/d} & p < d/2, \, q \neq d/(d-p) \end{cases}.$$

Then as per Theorem 1 of Fournier & Guillin (2015), there exists a constant $C(p, q, d)$ (a function of $p, q, d$) such that, $\mathbb{E}_{\mathbf{z} \sim \mu^{\otimes n}}$ from $(**)$ can be bounded from above by $C M_q^{p/q}(\mu) G(n)$. We obtain:

$$\sup_\mu \sup_x \int_{\Omega^n} \|\mathcal{F}(x, \mu) - \mathcal{F}(x, \nu_{\mathbf{z}}^n)\|_\infty \, d\mu^{\otimes n}(\mathbf{z}) \leq \mathscr{L} \sup_\mu C M_q^{p/q}(\mu) G(n)$$

$$\leq \mathscr{L} C \, diam(\Omega)^p G(n), \tag{22}$$

where we have used the fact that $\mu$ is a probability measure on $\Omega$.

Next, we obtain an upper bound for the second integral in $(*)$.

$$\sup_{\mu} \sup_{x} \int_{\Omega^n} \|\mathcal{F}(x, \nu_{\mathbf{z}}^n) - (T(x, \mathbf{z}))_1\|_{\infty} \, d\mu^{\otimes n}(\mathbf{z})$$

$$\leq \sup_{\mu} \int_{\Omega^n} \sup_{x} \|\mathcal{F}(x, \nu_{\mathbf{z}}^n) - (F_{n+1}(x, z_1, \ldots, z_n))_1\|_{\infty} \, d\mu^{\otimes n}(\mathbf{z})$$

$$+ \sup_{\mu} \int_{\Omega^n} \sup_{x} \|(F_{n+1}(x, z_1, \ldots, z_n))_1 - (T(x, \mathbf{z}))_1\|_{\infty} \, d\mu^{\otimes n}(\mathbf{z})$$

Consider the first term in the expression above

$$\sup_{\mu} \int_{\Omega^n} \sup_{x} \|\mathcal{F}(x, \nu_{\mathbf{z}}^n) - (F_{n+1}(x, z_1, \ldots, z_n))_1\|_{\infty} \, d\mu^{\otimes n}(\mathbf{z})$$

$$= \sup_{\mu} \int_{\Omega^n} \sup_{x} \left\| \mathcal{F}(x, \nu_{\mathbf{z}}^n) - \mathcal{F}\left(x, \nu_{(x,\mathbf{z})}^{n+1}\right) \right\|_{\infty} \, d\mu^{\otimes n}(\mathbf{z})$$

$$\leq \sup_{\mu} \int_{\Omega^n} \mathscr{L} \left\| \nu_{\mathbf{z}}^n - \nu_{(x,\mathbf{z})}^{n+1} \right\|_{\mathcal{W}_p} \, d\mu^{\otimes n}(\mathbf{z})$$

$$\leq \sup_{\mu} \int_{\Omega^n} \mathscr{L} \frac{2}{n+1} d\mu^{\otimes n}(\mathbf{z})$$

$$= \mathscr{L} \frac{2}{n+1} \tag{23}$$

Since we have assumed 16, we have

$$\sup_{\mu} \int_{\Omega^n} \sup_{x} \|(F_{n+1}(x, z_1, \ldots, z_n))_1 - (T(x, \mathbf{z}))_1\|_{\infty} \, d\mu^{\otimes n}(\mathbf{z}) \leq \varepsilon \tag{24}$$

Putting together 22, 23, and 24, we get that

$$\|\mathcal{T}_n - \mathcal{F}\| \leq \mathscr{L} C \operatorname{diam}(\Omega)^p G(n) + \mathscr{L} \frac{2}{n+1} + \varepsilon.$$

$\square$

## B    PROOF OF THEOREM 4

**Proposition 1.** *Suppose there exists a Lagrangian solution* $\mu \in C([0, \tau]; \mathcal{P}(\mathbb{R}^d))$ *of equation 2. Additionally, suppose that* $\mathcal{F}$ *satisfies Assumption 1. Then the solution satisfies,*

$$\operatorname{supp} \mu(t) \subseteq B_{C_t}(0) \tag{25}$$

*for all* $t \in [0, \tau]$, *where* $C_t = (R + 2\mathscr{M}t)e^{3\mathscr{M}t}$.

*Proof.* By definition of the Lagrangian solution,

$$\|X(t, x)\|_{L_1} \leq \|x\|_{L_1} + \int_0^t \|\mathcal{F}(X(s, x), \mu(s))\|_{L_1} ds \tag{26}$$

$$\leq \|x\|_{L_1} + \int_0^t \mathscr{M}(1 + \|X(s, x)\|_{L_1} + M_1(\mu(s))) ds \tag{27}$$

Integrating both sides of 27 with respect to $\mu_0$ we get,

$$M_1(\mu(t)) \leq M_1(\mu_0) + \mathscr{M} \int_0^t (1 + 2M_1(\mu(s))) ds$$

Combining this with 27 itself we get,

$$\|X(t, x)\|_{L_1} + M_1(\mu(t)) \leq M_1(\mu_0) + \mathscr{M} \int_0^t (2 + \|X(s, x)\|_{L_1} + 3M_1(\mu(s))) ds$$

Using Gronwall's lemma, this implies

$$\|X(t,x)\|_{L_1} + M_1(\mu(t)) \leq (M_1(\mu_0) + 2\mathscr{M}t))e^{3\mathscr{M}t}$$
$$\leq (R + 2\mathscr{M}t)e^{3\mathscr{M}t}$$

$\square$

**Theorem 4** (Mean Field Dynamics Approximation Using Transformers). *Let $\delta > 0$ and $n \geq 1$. Suppose $\mathcal{F}$ satisfies Assumption 1 for some $p$ and Assumption 2 for $\mu_0 \in \mathcal{P}_c(\mathbb{R}^d)$. If $\mathrm{supp}\,\mu_0 \subseteq B_R(0)$ for some $R > 0$, and $K \in \mathbb{R}$ is such that $K > (R + 2\mathscr{M}\tau e^{3(\mathscr{M}\tau)})$. $\mathcal{T}_n$ satisfies for all $z \in \bar{B}_K(0)$ and $\mu \in \mathcal{P}(\bar{B}_K(0))$*

$$\|\mathcal{T}_n(x,\mu) - \mathcal{F}(z,\mu)\|_{L_\infty} < \delta$$

*Then we have that*

$$W_p(\mu^{\mathcal{F}}(t), \mu^{\mathcal{T}_n}(t)) < \delta \cdot 2^p t \exp(2^p \mathscr{L}t) \tag{19}$$

*where $\mu^{\mathcal{F}}$ and $\mu^{\mathcal{T}_n}$ are the solutions to equation 2 and equation 10, respectively, and the constants are independent of $\mu_0 \in \mathcal{P}(B_R(0))$.*

*Proof.* Let $X, Y$ be the flow maps associated with with respect the vector fields $\mathcal{F}$ and $\mathcal{T}_n$, respectively. From the definition of Lagrangian solutions we know that,

$$X(t,x) = x + \int_0^t \mathcal{F}(X(s,x), \mu^{\mathcal{F}}(s))ds$$

$$Y(t,x) = x + \int_0^t \mathcal{T}_n(Y(s,x)), \mu^{\mathcal{T}_n}(s))ds$$

for all $t \in [0, \tau]$. From this we get

$$\|Y(t,x) - X(t,x)\|_p^p = \left\| \int_0^t \mathcal{T}_n(Y(s,x), \mu^{\mathcal{T}_n}(s))ds - \int_0^t \mathcal{F}(X(s,x), \mu^{\mathcal{F}}(s))ds \right\|_p^p$$

$$\leq 2^{p-1} \left\| \int_0^t \mathcal{T}_n(Y(s,x), \mu^{\mathcal{T}_n}(s))ds - \int_0^t \mathcal{F}(Y(s,x), \mu^{\mathcal{F}}(s))ds \right\|_p^p$$

$$+ 2^{p-1} \left\| \int_0^t \mathcal{F}(Y(s,x), \mu^{\mathcal{F}}(s))ds - \int_0^t \mathcal{F}(X(s,x), \mu^{\mathcal{F}}(s))ds \right\|_p^p$$

$$\leq 2^{p-1} \left\| \int_0^t \mathcal{T}_n(Y(s,x), \mu^{\mathcal{T}_n}(s))ds - \int_0^t \mathcal{F}(Y(s,x), \mu^{\mathcal{T}_n}(s))ds \right\|_p^p$$

$$+ 2^{p-1} \left\| \int_0^t \mathcal{F}(Y(s,x), \mu^{\mathcal{T}_n}(s))ds - \int_0^t \mathcal{F}(Y(s,x), \mu^{\mathcal{F}}(s))ds \right\|_p^p$$

$$+ 2^{p-1} \left\| \int_0^t \mathcal{F}(Y(s,x), \mu^{\mathcal{F}}(s))ds - \int_0^t \mathcal{F}(X(s,x), \mu^{\mathcal{F}}(s))ds \right\|_p^p$$

The first inequality follows from Young's inequality. Since, $K > (R + 2\mathscr{M}\tau e^{3(\mathscr{M}\tau)})$, for $\varepsilon > 0$ small enough, and $n$ large enough we can conclude that, $K > (R + 2(\mathscr{M} + \varepsilon + \delta(n,K))\tau e^{3((\mathscr{M}+\varepsilon+\delta(n,K))\tau)})$. We can approximate $\mathcal{F}$ using $\mathcal{T}_n$ on $B_K(0)$. Using the uniform norm approximation, we can conclude that

$$\|\mathcal{T}_n(x,\mu)\|_{L_1} \leq (\mathscr{M} + \varepsilon + \delta(n,K))(1 + \|x\|_{L_1} + M_1(\mu))$$

for all $x \in \mathbb{R}^d$ and $\mu \in \mathcal{P}(B_K(0))$. From this we get,

$$\mathrm{supp}\,\mu^{\mathcal{T}_n(s)} \subseteq B_K(0)$$

by Proposition 1.

$$2^{1-p}\|Y(t,x) - X(t,x)\|_p^p \leq \int_0^t \varepsilon + \delta(n,K)ds$$

$$+ \int_0^t \mathscr{L}\mathcal{W}_p^p(\mu^{\mathcal{F}}(s), \mu^{\mathcal{T}_n}(s))ds + \int_0^t \mathscr{L}\|Y(s,x) - X(s,x)\|_{L_1}ds.$$

Taking expectation with respect to the initial condition we obtain,

$$2^{1-p} \int_{\mathbb{R}^d} \| \|Y(t,x) - X(t,x)\|_p^p d\mu_0(x) \le (\varepsilon + \delta(n,K))t + \int_0^t \mathscr{L} \mathcal{W}_1(\mu^{\mathcal{F}}(s), \mu^{\mathcal{T}_n}(s)) ds$$

$$+ \int_{\mathbb{R}^d} \int_0^t \mathscr{L} \|Y(s,x) - X(s,x)\|_p ds \, d\mu_0(x)$$

Using the fact that

$$\mathcal{W}_p(\mu^{\mathcal{F}}(t), \mu^{\mathcal{T}_n}(t)) \le \int_{\mathbb{R}^d} \|x - X(t, Y^{-1}(t,x))\|_p^p d(Y(t,\cdot)_{\#}\mu_0)(x)$$

$$= \int_{\mathbb{R}^d} \|Y(t,x) - X(t,x)\|_p^p d\mu_0(x),$$

we can conclude that

$$2^{-p} \int_{\mathbb{R}^d} \|Y(t,x) - X(t,x)\|_p^p d\mu_0(x) + 2^{-p} \mathcal{W}_p^p(\mu^{\mathcal{F}}(t), \mu^{\mathcal{T}_n}(t)) \le (\varepsilon + \delta(n,K))t$$

$$+ \int_0^t \mathscr{L} \mathcal{W}_p^p(\mu^{\mathcal{F}}(s), \mu^{\mathcal{T}_n}(s)) ds + \int_{\mathbb{R}^d} \int_0^t \mathscr{L} |Y(s,x) - X(s,x)| ds \, d\mu_0(x).$$

This implies that

$$2^{-p} \int_{\mathbb{R}^d} \|Y(t,x) - X(t,x)\|_p^p d\mu_0(x) + 2^{-p} \mathcal{W}_p^p(\mu^{\mathcal{F}}(t), \mu^{\mathcal{T}_n}(t)) \le (\varepsilon + \delta(n,K))t$$

$$+ \mathscr{L} \int_0^t \mathcal{W}_p^p(\mu^{\mathcal{F}}(s), \mu^{\mathcal{T}_n}(s)) ds + \mathscr{L} \int_{\mathbb{R}^d} \int_0^t |Y(s,x) - X(s,x)| ds \, d\mu_0(x)$$

Now, applying Gronwall's inequality, we get,

$$\int_{\mathbb{R}^d} \|Y(t,x) - X(t,x)\|_p^p d\mu_0(x) + \mathcal{W}_p^p(\mu^{\mathcal{F}}(t), \mu^{\mathcal{T}_n}(t)) \le 2^p(\varepsilon + \delta(n,K))t \exp(2\mathscr{L}t)$$

This implies that

$$\mathcal{W}_p^p(\mu^{\mathcal{F}}(t), \mu^{\mathcal{T}_n}(t)) \le 2^p(\varepsilon + \delta(n,K))t \exp(2\mathscr{L}t)$$

$\square$

