# OpenReview forum: "Identification of Mean-Field Dynamics using Transformers"
_ICLR.cc/2025/Conference — ICLR 2025 Conference Withdrawn Submission_

### Official Review · Reviewer_YKsa · 2024-10-29

**Soundness:** 2
**Presentation:** 2
**Contribution:** 2
**Rating:** 3
**Confidence:** 3

**Summary:**

A transformer-variant tailored to vector fields as appearing in mean field dynamical systems is proposed. Based on established approximation results for transformers and mean field dynamics, approximation results for the vector fields and the resulting dynamics are shown. The model is illustrated with two simple numerical experiments.

**Strengths:**

* Novel use of transformers in the context of mean field dynamical systems
* At least qualitative approximation results are available

**Weaknesses:**

* Small technical contribution: Straightforward modification of the transformer model, standard approximation results
* In the present form, rather narrow field of application (though more applications could be found, since mean field dynamics are ubiquitous)
* Insufficient experiments, both more (and more detailed) experiments as well as a better investigation of the outcomes are necessary. For example
  * More settings (e.g., what if a Cucker-Smale system does not reach consensus? Is the proposed method able to deal with these different asymptotic behaviours?)
  * More models (e.g., what about Cucker-Dong or Hegelsmann-Krause models?)
  * Higher dimensions (e.g., high-dimensional Cucker-Smale systems)
* Similarly, more details could be provided regarding the outcomes of the experiment. How reliable are the learned models? What about generalization (e.g., to different asymptotic behaviors)? What about identifiability of the underlying systems?

* Presentation needs to be improved, multiple typos, language issues and imprecisions. Some examples are given **below**.



L029: Language

Related work L042-44: Feng et al '21, Lu et al '19 work on the microscopic level, not the mean field level

L055: Ref should appear earlier

L061: A reference for the continuity equation should be added here, since this is a field usually not known in machine learning

L094: Why is this reference relevant?

L149: What about existence and uniqueness?

Equ (5): There is no ReLU here

L190: Output shape of $T$ is ambiguous

L200: Language

Equ (8): For consistent notation, parameters should be included here

Section title of 3.2: "with" instead of "of"

L228: Language

L234: Wrong inequality direction

L261: Language

Equ (12): Formatting

L326: Typo

L390: Is this a condition or a consequence?

L423: Language

L471: Superfluous "."

Section, Cucker Smale: How are the initial velocities computed?

Section 5.1: What does "trained with SGD" means here? And how is the transformer model then trained if not with a variant of SGD?

L505: "Note that the transformer model does not compute any gradients." This is at odds with the description before Section 5.1

Figure 1: Formatting problem

L534: Language

**Questions:**

1. The experiments need to be significantly expanded. Furthermore, both the description of the experiments as well as of the results needs to be considerably improved. Can you comment?
2. The title of this work contains "IDENTIFICATION OF MEAN-FIELD DYNAMICS USING TRANSFORMERS", but the setup is more about prediction of trajectories. Can you perform additional experiments that show that the vector field is reliable estimated? Can this be already observed in the two experiments in the current version (difficult to infer from the rather short description)?
3. Is it possible with this approach to arrive at uniform-in-time approximation guarantees? With the current strategy it does not work.

---

### Official Review · Reviewer_g1gp · 2024-10-30

**Soundness:** 1
**Presentation:** 2
**Contribution:** 2
**Rating:** 3
**Confidence:** 4

**Summary:**

The paper "Identification of Mean-Field Dynamics Using Transformers" introduces a continuum version of the transformer architectures to approximate the mean-field dynamics in interacting particle systems, commonly seen in physics, biology, and engineering. The authors show that based on the fact that a  finite-dimensional transformer can effectively approximate a finite dimensional vector, their "expected transformer," defined as an expectation of finite-dimensional transformers,  approximates the infinite-dimensional mean-field dynamics well. The authors present theoretical results for approximation bounds, as well as numerical simulations to validate its effectiveness. Their study shows that transformers, with their permutation-equivariant properties, can be used for modeling complex, collective behaviors in particle systems.

**Strengths:**

The authors introduce a new application for transformer architectures in modeling mean-field dynamics which is interesting as it shows new potential for transformers beyond their usual scope of traditional NLP and image tasks. The paper presents a strong theoretical foundation for approximation bounds of the architecture based on the universal approximation theorem and compares it with related work. The suggested 'expected transformers' is then applied to two systems, Cucker-Smale and the training dynamics of a two-layered network, to complement the theoretical results.

**Weaknesses:**

Unfortunately, the Reviewer can not recommend the paper in its current form for publication due to some significant weaknesses:

- The paper's overall positioning is unclear. Although submitted in the learning theory category and featuring training dynamics of two-layer networks as an example, the introduction and motivation focus extensively on "identifying dynamical system models for physical processes." This approach suggests an application-focused paper aimed at modeling systems with many particles or agents. However, the related work section then shifts to reviewing literature on PINNs, Neural Operators, and methods for approximating differential equations. To improve clarity, the authors could either position the paper as primarily application-driven, aiming to model particle/agent systems, or, if the focus is indeed on learning theory, adjust the introduction and related work to align with this emphasis.

- From the reviewers perspective, the use of the theoretical results based on the universal approximation theorem in practice is very limited as they do not provide error rates. The authors also do not refer to their own theoretical results at all in the numerical illustration section. The reviewer appreciates that the authors acknowledge such limitations at the beginning of section 5, but this leads to no real connection between the theoretical and experimental parts of the paper.

- The "Numerical Simulations" section requires more precision and clarity. Besides the ambiguous title, the results and their interpretations are inconclusive. For example, Figure 1 shows that the training and test losses for the SGD approach are significantly lower, contradicting the authors' claim that "the transformer-trained models exhibit favorable training and test loss performance," which seems questionable. Furthermore, the figure on page 9 is partially cut off due to formatting issues. Descriptions should also be refined, replacing vague terms like "quite small" and "quite well" with precise, quantitative evaluations.

- The reviewer would suggest to report the details of the architecture used and conduct ablation studies to offer valuable insights into the model's performance.

**Questions:**

See Weaknesses and additionally:

- Theorem 2: The parameters $\theta_j$ and $\zeta_j$ depend linearly on the dimension d (line 244). Does this refer to the number of parameters  ? Moreover, the $\zeta$ are the parameters of the fully connected layers. The cited reference states that there is no explicit control over the dependency of the number of MLP parameters $\zeta$ on $\epsilon$. Are the authors considering a special case that allows for bounds on $\zeta$ ?

---

### Official Review · Reviewer_E9Kx · 2024-10-30

**Soundness:** 1
**Presentation:** 1
**Contribution:** 2
**Rating:** 3
**Confidence:** 5

**Summary:**

This paper proposes a method to approximate mean-field dynamics of interacting particle systems using transformers. It leverages the fact that such dynamics are permutation-equivalent.

**Strengths:**

The main strength of this paper is its intriguing idea to use permutation-equivariant transformers to approximate infinite-dimensional solutions to dynamical systems, which are typically challenging to approximate due to mean-field interactions.

**Weaknesses:**

While the idea is interesting, the theoretical contribution appears limited and unsound, and the simulations are questionable and difficult to reproduce.

To be more specific, here is a list of some of my concerns:
1. The theoretical contributions of this paper are limited and poorly presented. Theorem 3 and Corollary 1 are straightforward consequences of Alberti et al. and a standard result on the approximation of measures. Additionally, it is unclear whether (16) should hold for all $n$ (as $n$ is refered before being defined). I have two suggestions here: a) $n$ should be defined earlier so that the right-hand side in line 317 is small, with the transformer defined afterwards; b) Theorem 3 and Corollary 1 should be merged, as the only difference between them is whether the result by Alberti et al. is cited.
2. The paper aims to propose a method for approximating solutions to mean-field systems, yet no practical method is actually studied. Indeed, $\mathcal{T}_n$ cannot be computed as it involves an expectation, and $\mu^{\mathcal{T}_n}$ cannot be computed either, as it requires applying $\mathcal{T}_n$ multiple times.  This raises important questions not addressed in the paper, such as: 'In practice, should the same trajectories be used at each time step when computing $\mu^{\mathcal{T}_n}$?". The paper would benefit from performing the theoretical analysis on the practical method proposed in Remark 1 rather than on $\mathcal{T}_n$. Furthermore, a practical method for approximating $\mu^{\mathcal{F}}$ should be proposed and studied in place of $\mu^{\mathcal{T}_n}$.
3. Theorem 4 assumes that $\mathcal{F}$ satisfies Assumption 2 while it should be $\mathcal{T}_n$,  which is not defined at that point. Consequently, both Assumption 2 and Theorem 4 lack clarity as they stand. In my view, Assumption 2 should be replaced with: "for any $n$ and for $\mathcal{T}_n$ constructed from a transformer $T$, (18) should have a unique Lagrangian solution". Additionally, the contribution of this paper would be strengthened if such a property were proven rather than assumed. At the very least, the authors should explain the difficulties in proving such results; otherwise, it leaves the impression that they did not attempt to do so.
4. Many of the norms used in the paper are imprecise or incorrect in terms of space, including those on lines 235, 243, 269, 271, 284, 317, 350, etc.
5. Figure 3 does not appear in the correct position.
6. The description of the simulations is too brief. As it is now, the simulations are not reproducible to me.
7. Some notations are used multiple times for different objects, making the article more difficult to read.
8. There are a large number of typos.
9. A few words on the computational complexities of the proposed methods would strengthen the analysis.

**Questions:**

Here are my questions :
1. In Remark 1, the authors divide the data into $B$ batches of $n$ samples. Can they discuss more this choice? On the one hand, taking $B=1$ seems the best choice for approximating the mean-field dynamic. On the other hand, the computational cost is linear in $B$ and quadratic in $n$ so that large $n$ are not preferable. What kind of trade-off could the authors propose?
2. Why is the analysis done on $\mathcal{T}_n$ and not on the the practical algorithm described in Remark 1?
3. Can the authors propose a practical method to approximate $\mu^{\mathcal{F}}$? How would Theorem 4 adapt to such a practical method?
4. Why does SGD works better than transformers on figure 1? Would increasing $N$ reduce this difference?
5. On Cucker-smale, taking $N=20$ seems very small, can the author comment? Would it be preferable to solve Cucker-Small using a PDE solver on the mean-field equations?

---

### Official Review · Reviewer_7Rab · 2024-11-05

**Soundness:** 3
**Presentation:** 3
**Contribution:** 2
**Rating:** 5
**Confidence:** 4

**Summary:**

In this paper, the authors proved the approximation results to identify a coupled particle system and its mean field equations by certain transformer architectures. This work is within the scope of theory  based on the universal approximation theorem of transformer ( Alberti et al. (2023)). They presented numerical examples to demonstrate the efficiency of the theoretical results.  In my opinion, the contribution of the present paper is technical. And the mathematical parts as far as I have checked are correct.

**Strengths:**

This paper gave a view of investigating collective dynamics of coupled  systems of many particle by the machine learning approach, in particular, the transformer-based models, and its theoretical foundation of universal approximation.

**Weaknesses:**

In my opinion, I think the novelty and technical contribution of the present work is not very significant. It is mainly based on the existing transformer approximation theorem. The assumptions of Theorems (eq. 16) and theorem 4 (the inequality above (19)) seem too strong, which should be further validated.

**Questions:**

1. How do the assumptions of Theorems (eq. 16) and theorem 4 (the inequality above (19)) hold? Theses core conditions should be validated.
2. How do the global Lipchitz and linear growth conditions  hold? Can we validate for the Cuker-Smale system in the example?
3. It is not very clear to me which architectures of transformer models were used to approximate the numerical examples?

---

### Note · Authors · 2024-11-12

**Comment:**

We thank the reviewers for their comments. We shall use their feedback to improve the paper.

**Withdrawal Confirmation:**

I have read and agree with the venue's withdrawal policy on behalf of myself and my co-authors.